# Secondary Anterior Cruciate Ligament Injury Prevention Training in Athletes: What Is the Missing Link?

**DOI:** 10.3390/ijerph20064821

**Published:** 2023-03-09

**Authors:** Choi-Yan (Tiffany) Wong, Kam-Ming Mok, Shu-Hang (Patrick) Yung

**Affiliations:** 1Department of Orthopaedics & Traumatology, Faculty of Medicine, The Chinese University of Hong Kong, Hong Kong, China; 1155110993@link.cuhk.edu.hk (C.-Y.W.); kmmok2@ln.edu.hk (K.-M.M.); 2Office of Student Affairs, Lingnan University, Hong Kong, China; 3School of Interdisciplinary Studies, Lingnan University, Hong Kong, China

**Keywords:** sports medicine, knee, physical wellness, physical activity

## Abstract

After reconstruction, the return to full competition rate of athletes is low, while the re-injury rate remains high despite the completion of a rehabilitation programme. Primary ACL prevention programmes are well developed, yet few research papers focus on secondary ACL injury prevention. The aim of current review is to determine if current ACL secondary prevention training has a positive influence on the re-injury rate, the clinical or functional outcomes, or the risk of re-injury in athletes. Studies investigating secondary prevention of ACL were searched in PubMed and EBSCOhost, followed by a review of the references in the identified articles. The existing evidence suggests that neuromuscular training, eccentric strengthening, and plyometric exercises may have a potential impact on improving biomechanical, functional, and psychological outcomes in athletes; however, the studies on the prevention of second ACL injury in athletes is scarce and inconclusive. Future research is needed to investigate the effectiveness of secondary ACL prevention in reducing the re-injury rates. (PROSPERO Registration number: CRD42021291308).

## 1. Introduction

Anterior Cruciate Ligament (ACL) injuries are one of the most common traumatic knee injuries in sports [1]. Approximately 200,000 to 250,000 ACL injuries occur annually within the United States [2], which have doubled over the past two decades despite the increasing research effort and the development of ACL prevention programmes [3]. ACL injuries may often be associated with meniscus injuries (55–65%) and cartilage injuries (16–46%) [4], causing a four-times higher risk of knee osteoarthritis [5], where patients may end up requiring a total knee replacement [6]. ACL injuries may also lead to an impairment of knee-related quality of life at 5 to 25 years [7].

Approximately 175,000 ACL reconstructions (ACLR) are performed each year in the United States [8]. After one year of surgery, 66% of the athletes were able to participate in a modified or full competition [9], and 55% could return to competition at their pre-injury level [10]. However, athletes who return to high levels of sport are 30 to 40 times more likely to suffer a second ACL injury compared with uninjured athlete [11]. It is reported that nearly 1 in 3 to 4 young, active athletes will sustain an ipsilateral or contralateral ACL injury after returning to sports [12], while nearly half of those happen within two months of returning to sports [13]. It was reported in a previous systematic review that the overall rate of secondary ACL injury was 27% in young athletes after ACLR [14], while young female athletes may have an even higher rate of re-injury up to 32% [15,16].

Previous ACL injuries not only contribute to a higher risk of ipsilateral ACL re-injury but also of contralateral ACL injuries [17]. The risk of a contralateral ACL injury was reported to be 5% in male athletes and 26% in female athletes, which means the risk of female athletes suffering from a contralateral ACL injury is as high as six times compared to male athletes [18].

The factors contributing to a higher risk of a second ACL injury include limb asymmetry in muscle strength and functional performance [19], neuromuscular impairments [20], psychological factors such as fear of re-injury and poor self-efficacy [21], and proprioceptive loss [22]. Quadriceps femoris muscle strength asymmetries may be associated with gait asymmetries, impaired alignment of the hip and knee, and alteration in knee biomechanics [23]. The altered neuromuscular timing and recruitment can lead to dynamic knee valgus stress in the lower limb; it has also been reported that female athletes show four times greater activation of their hamstring muscles than males during knee dynamic stress motion [24]. As muscle strength, neuromuscular control, and joint proprioception are essential factors contributing to the dynamic stability of the knee joint and are much impaired after an ACL reconstruction [22], exercises or training may be required to restore knee stability and function after a standardised rehabilitation programme.

Multiple reviews and meta-analyses on ACL injury prevention programmes (IPPs) have found that both plyometric and strength exercises were effective in preventing primary ACL injuries, while mixed results were found as to whether or not balance training needs to be included [25,26,27]. However, there is a lack of reviews or evidence towards secondary ACL injury prevention despite a low return to full competition rate and a high re-injury rate, as mentioned above.

The current review aims to provide insights into the impacts of currently developed ACL secondary injury prevention training on athletes. This may include the effectiveness to reduce a secondary injury rate or to reduce the risk of re-injury, types, and intensity of exercises that may be beneficial, and the ability to restore a normal knee function or better clinical and functional outcomes. These may help to (1) improve the understanding of how current secondary ACL prevention training is developed; (2) improve the understanding of the effectiveness of different types of exercises in reducing the risk of re-injury; (3) improve athletic performance while reducing the re-injury rate when athletes return to competitive sports.

## 2. Materials and Methods

Data management of this systematic review is reported in line with the Preferred Reporting Items for Systematic Review and Meta-Analyses (PRISMA) [28]. The review protocol has been registered in PROSPERO (Registration number: CRD42021291308).

### 2.1. Search Strategy

An electronic literature search has been conducted of the PubMed (1964 to 2021) and EBSCOhost (CINAHL, MEDLINE, ScienceDirect (1985 to 2021)) databases. The search identified all articles containing the terms “ACL” or “Anterior cruciate ligament”, “Second* injur* prevent*” or (“Reinjur* or re-injur* or recur*” and prevent*) or (Reconstruct* and train*), and “Athlet*.

### 2.2. Selection

Randomised control trials, randomised clinical trials, control trials, and therapeutic studies have been included. Review articles and meta-analyses have been included initially to locate all possible related studies. The reference lists of the included articles have also been reviewed for relevant articles. Dissertations, textbook chapters, articles without a full copy, qualitative studies, literature reviews, guidelines, audits, and single case studies were not considered.

The inclusion criteria included (1) English-language studies, (2) studies that contain a secondary ACL injury prevention training for athletes who have undergone a unilateral ACL reconstruction, (3) subjects included had a primary unilateral ACL injury and have completed an ACL reconstruction rehabilitation, and (4) studies that contain the effects of training on the clinical or functional outcomes, or modification of ACL re-injury risk factors, or influence on the rate of an ipsilateral or contralateral ACL re-injury. The exclusion criteria included (1) studies of primary ACL injury prevention, and (2) the subjects included had any previous ACL injuries on either limb with or without a reconstruction before this incidence.

### 2.3. Data Analysis

The articles identified were assessed for inclusion according to the inclusion criteria mentioned above based on their title and abstract according to an abstract review form. If they met the criteria or if it was unclear, the full text was retrieved. Irrelevant studies were excluded. The data extraction and the risk of bias assessment were completed with reference to the data extraction form provided by Cochrane Developmental, Psychosocial and Learning Problems, 2014 [29]. The data from each included study were abstracted for (1) sample demographics and sample size; (2) any control or comparison groups; (3) the intervention; (4) any information about the ACLR and post-ACLR rehabilitation; (5) ACL secondary prevention training components, duration, and the exercise types; (6) the key outcome measures; and (7) the key findings after the ACL secondary injury prevention training.

## 3. Results

The literature search elicited a total of 1581 references, while 345 were found to be duplicated and another 1212 were excluded. Two additional articles were identified through a second-hand search of articles eligible. A total of 26 studies were reviewed in full-text, while 16 were excluded. The risk of bias was assessed using the Cochrane risk of bias framework. Nine articles met the final inclusion criteria for the current systematic review [20,30,31,32,33,34,35,36,37] (Figure 1).

Of the nine studies, none investigated the effects of secondary prevention programmes on the ACL re-injury rate in athletes, while all of them assessed the influence of these programmes on the modification of risk factors for a second ACL injury in athletes. All nine studies were prospective, while six were prospective randomised controlled trials, and three were prospective randomised clinical trials. Two studies were conducted for elite athletes at provincial or international levels. Table 1 summarises the characteristics of the participants, the surgical and rehabilitation information, the intervention, the key outcomes, and the key findings of the nine studies. These studies involved a total of 347 subjects, who were active in sports participation or competition before the injury. Overall, four studies found that secondary prevention programmes (including neuromuscular training, eccentric, and plyometric exercises) were effective in modifying risk factors for re-injury in athletes, including improving knee proprioception, functional performance, knee stability, landing biomechanics, and psychological readiness to return to sports. In contrast, five studies found that secondary prevention programmes (including running retraining, strength, agility, plyometrics, and perturbation training) were not effective in modifying the risk factors of an ACL re-injury in athletes (three focusing on restoring gait symmetries, one focusing on muscular and functional recovery, and one focusing on functional outcomes). Table 2 summarises the risk of bias in the included studies according to the Cochrane risk of bias framework [29].

## 4. Discussion

The aim of the current review is to determine if current ACL secondary prevention training has a positive influence on the re-injury rate, the clinical or functional outcomes, or the risk of re-injury in athletes. The principal findings of the current systematic review are as follows: (1) the available literature related to secondary ACL prevention in athletes is scarce; (2) the level of evidence of the existing literature is diversified; (3) most of the studies investigated if secondary prevention may be effective in modifying the risk factors of ACL re-injury rather than to reduce the re-injury rates; (4) the effectiveness of secondary prevention on modifying the risk factors of ACL re-injury in athletes is controversial; (5) the existing evidence suggests that neuromuscular training, eccentric strengthening, and plyometric exercises may improve biomechanical, functional, and psychological outcomes, and knee proprioception in athletes after ACLR and rehabilitation, which is similar to the current findings of a primary ACL prevention programme; and (6) there is no evidence that a secondary injury prevention program can accelerate the regaining of gait symmetry after ACLR.

The existing literature mainly focuses on the risk factors that contribute to an ACL injury, primary injury prevention programmes, surgical techniques, and rehabilitation after ACLR [38]. The available literature related to secondary ACL prevention in athletes is scarce, while the quality and level of evidence are diversified. There are limited randomised trials available, while most literature available are cohort studies or pilot studies, with a lower level of evidence (level II or III evidence) [39,40,41,42,43,44,45,46]. As cohort studies or pilot studies may potentially have a higher risk of bias compared with randomised trials, it would be difficult to interpret and evaluate the effects of interventions with randomised trials [47]. Therefore, the current review included only randomised trials (level I or II evidence).

In addition, most existing literature reported the influence of secondary prevention on the modification of the risk factors of ACL re-injury rather than the re-injury rates. Two papers reported the re-injury rates after participating in a secondary ACL prevention programme [48,49]. However, both papers were secondary analyses and were excluded from the current review. In the nine included studies, there was no report of the re-injury rate. As the modification of ACL re-injury risk factors may not directly relate to the re-injury rate or the risk of re-injury, more research is needed in the future to investigate the effectiveness of secondary ACL prevention on reducing the re-injury rate.

In the nine included studies, the risk factors they aimed to modify and the exercise approach they used varied from one another. Therefore, the effectiveness of secondary prevention on modifying the risk factors of ACL re-injury is controversial, depending on the specific risk factors they were looking into, and the exercise approaches they used. The risk factors that the included studies focused on were (1) impaired neuromuscular control, knee stability, and proprioception loss; (2) asymmetry in muscle strength and functional performance; (3) psychological response; and (4) gait asymmetry.

Six studies looked into neuromuscular control, knee stability, proprioception, and muscular and functional recovery [20,31,34,35,36,37]. One of the six studies investigated if the clinical and functional outcomes could be different for athletes participating in a secondary prevention programme or a secondary prevention programme plus perturbation training [20]. They found no clinically meaningful differences between the two groups, but there was no report on the effectiveness of either group on improving the clinical or functional outcomes. Another study found that a running retraining programme did not appear to influence the knee’s muscular and functional recovery [37]. The random sequence generation, allocation concealment, and blinding of outcome assessment were not documented in the above study, so the risk of bias was unclear. The remaining four studies reported the positive impacts of neuromuscular training towards neuromuscular control, knee dynamic stability, proprioception, and muscular and functional recovery [31,34,35,36], while Kasmi et al. (2021) reported a combination of eccentric and plyometric exercises may be the most effective to stimulate positive changes in the above outcomes. This may indicate that a specific neuromuscular training including strengthening and plyometric exercises may be essential to promote a positive neuromuscular and functional recovery, while solely a sports-specific training itself may be insufficient in promoting neuromuscular and functional improvement in athletes after ACLR and a traditional rehabilitation.

Regarding psychological responses, factors such as fear of re-injury, confidence, self-efficacy, and psychological readiness to return to sports were potential barriers for athletes after ACLR [50]. One of the nine studies mentioned that eccentric and plyometric exercises were able to improve the psychological outcomes [35]. They found that the combination of the two types of exercises was able to induce a greater improvement in the psychological status and level of confidence of athletes. However, the randomisation process may be potentially biased as the group allocation was realised by adjusting the BMI, age, and sex of the study participants. Multiple meta-analyses have found that strengthening and plyometric exercises were effective in preventing primary ACL injuries [25,26,27]. The current review suggested that these exercises may not only be effective in preventing a primary ACL injury, but also effective in influencing the risk factors of a re-injury, such as limb asymmetries, muscle imbalance, impaired proprioception and knee position sense, and psychological factors. In other words, a primary prevention programme may also be used for a secondary prevention purpose, especially when secondary prevention programmes are not widely developed nowadays.

Regarding gait asymmetries, three studies investigated if a secondary prevention programme or a secondary prevention programme plus perturbation training is effective in improving walking mechanics [30,32,33]. All of the three studies used a programme called ACL-SPORTS, which included muscle strengthening exercises, agility drills, balance exercises, dynamic sport-related tasks, and perturbation training. They all found that both groups were not effective in restoring gait symmetry in the short term, while gait asymmetries mostly resolved after 2 years post-ACLR, regardless of the intervention group. It is worth noting that the potential risk of bias may be high in Capin et al. (2017), as the random sequence generation, allocation concealment, and blinding procedures were not documented. As gait impairments may be present even in the absence of clinical or functional deficits [30,51,52], gait symmetries may not be a strong indicator or risk factor of an ACL re-injury. As gait may vary from time to time, gait assessment may not be as reliable and functional as other outcome measures, such as functional tests.

The primary strength of the current review is that the level of evidence is high as only randomised trials were included. This may potentially reduce the risk of bias. In addition, there is confidence that all adequate studies have been identified because the reference lists of all included and excluded studies have been screened or searched.

The current review also has some limitations. To start with, the participants of the included studies tended to share particular characteristics (very active in sports participation before the injury, achieved good recovery over the ACLR limb, a high percentage of muscle strength symmetry, and participating in sports involving frequent cutting and pivoting). In addition, the number of male and female athletes was unequal in the included studies (253 males, 94 females). Therefore, the results of the current review may not be generalisable to other populations such as athletes who have poor recovery and a higher extent of limb asymmetries after ACLR. Another limitation of the current review is that the ACL secondary prevention programmes included in the study are not sports specific. This is mainly limited by the scarcity of literature available in this field.

## 5. Conclusions

Given the scarcity of the literature available related to secondary ACL prevention in athletes with the diversified level of evidence and lack of reports towards the re-injury rate, more research is needed in the future regarding the effectiveness of secondary ACL prevention on reducing the re-injury rates. In addition, more research should be conducted investigating if sports-specific secondary prevention programmes may positively influence the ACL re-injury rates and modify the specific risk factors of re-injury in different sports. The effectiveness of secondary prevention in modifying the risk factors of ACL re-injury in athletes is controversial. The existing evidence suggests that neuromuscular training, eccentric strengthening, and plyometric exercises may have a positive impact on the biomechanical, functional, and psychological outcomes in athletes, while secondary prevention programmes may be ineffective in influencing gait asymmetries. This may indicate that a primary ACL prevention programme may also be valid in secondary ACL injury prevention.

## Figures and Tables

**Figure 1 ijerph-20-04821-f001:**
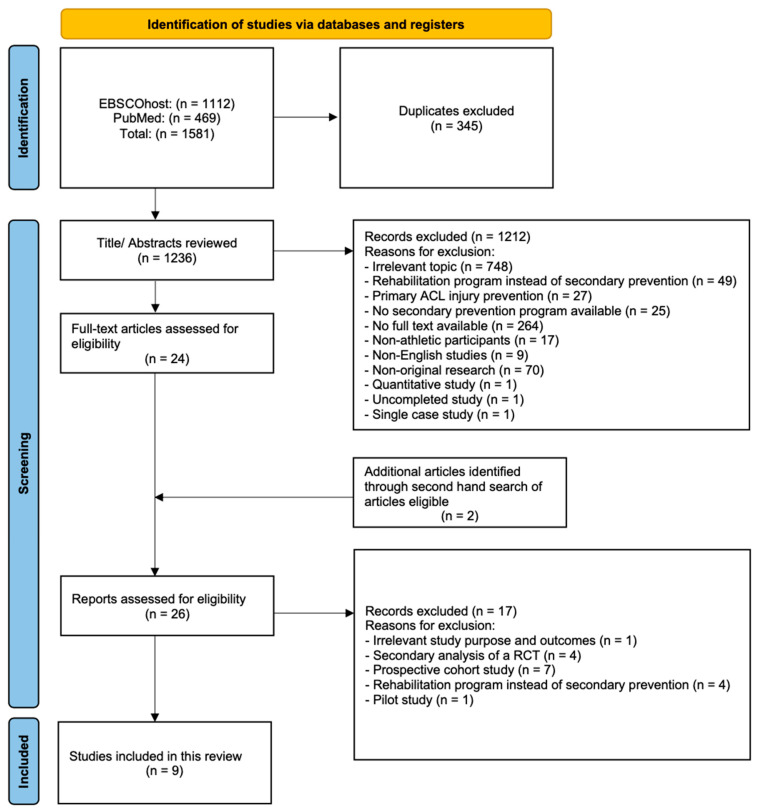
Flow of information through the systematic literature search and screening.

**Table 1 ijerph-20-04821-t001:** Summary of the included studies.

Study	Study Design	Intervention	Description of Participants	ACLR and Post-ACLR Rehabilitation	Intervention Description	Key Outcome Measures	Key Findings
[20]	Randomised clinical trial	Compare clinical and functional outcomes of SAP group (strength, agility, plyometric, and secondary prevention treatment) and SAP + PERT group (SAP + perturbation training).	N: 40 (20 × SAP, 20 × SAP + PERT) Male athletes (mean age ± SD at surgery 23 ± 7 years) Participated in level I or II sports ≥50 h/year before their injuries ≥80% quadriceps femoris muscle strength symmetry Minimal knee joint effusion Full ROM No reports of pain Able to complete a running progression	Rehabilitation not standardised Have completed outpatient rehabilitation Between 3–9 months after ACLR	Duration: 4 weeks Frequency: 2 times a week Type of exercise: SAP: Progressive secondary ACL injury prevention exercises, agility drills, balance, dynamic sport-related tasks, and muscle strengthening exercises. SAP + PERT: All exercises in SAP group augmented with perturbation training.	Quadriceps strength and single-legged hop limb symmetry self-reported knee scores (IKDC, KOOS-sports and recreation subscale, KOOS-quality-of-life subscale) The proportion who achieved self-reported normal knee function The time from surgery to passing return to sport criteria	There were no clinically meaningful differences between groups in knee function and self-reported outcome measures. The results indicate that perturbation training may not contribute additional benefit to knee functional or clinical outcomes.
[30]	Randomised clinical trial	Compare SAP group and SAP + PERT group with respect to gait mechanics and elimination of gait asymmetries 1 and 2 years after ACLR.	N: 40 (20 × SAP, 20 × SAP + PERT) Male athletes (mean age ± SD at surgery 23 ± 7 years) Participated in level I or II sports ≥50 h/year before their injuries ≥80% quadriceps femoris muscle strength symmetry Minimal knee joint effusion Full ROM No reports of pain Able to complete a running progression	≥12 weeks after ACLR	Duration: 4 weeks Frequency: 2 times a week Type of exercise: SAP: Progressive secondary ACL injury prevention exercises, agility drills, and plyometric exercises. SAP + PERT: All exercises in SAP group plus perturbation training (neuromuscular training requiring selective muscle activation in response to surface perturbations applied by a physical therapist).	Sagittal and frontal plane hip and knee angles and moments at peak knee flexion angle (pKFA) and peak knee extension angle (pKExtA) Sagittal plane hip and knee excursion during weight acceptance (ie, pKFA—initial contact) and midstance (i.e., pKExtA—pKFA)	Both groups were not effective in restoring interlimb symmetry among men 1 or 2 years after ACLR. Although gait asymmetries improved from 1 to 2 years postoperatively, meaningful asymmetries persisted in both groups.
[31]	Randomised clinical trial	Examine the effects of a neuromuscular training program that emphasizes external focus of attention cuing on biomechanics, knee proprioception, and patient-reported function.	N: 24 (12 × experimental group; 12 × control group) Male athletes Participate in sports involving frequent landing and cutting No joint effusion Pain-free knee active ROM Single leg hopping test completed without pain at an equivalent distance/rate of at least 80% of the contralateral limb	Between 6–12 months after ACLR Received unilateral hamstring tendon autograft ACLR performed by the same surgeon	Experimental group: Duration: 8 weeks Frequency: 3 times per week for week 1–6 and 2 times per week for week 7–8 Exercises included: double-leg squats, walking lunges, single-leg squats, double-leg drop jumps, single-leg stance on an unstable surface, single-leg standing long jumps. Control group: Duration: 8 weeks Content: Complete routine sport-specific skills training	Biomechanical testing (kinematic and kinetic data from the single-leg landing trials) Knee joint position sense Patient-reported function (IKDC)	The experimental group demonstrated improvements in landing biomechanics, proprioception, and patient-reported function. The control group demonstrated no changes in any variable over the same period.
[32]	Prospective randomised control trial	Compare SAP group and SAP + PERT group with respect to improvements in movement symmetry during walking.	N: 40 (20 × SAP, 20 × SAP + PERT) Female athletes Participated in level I or II sports ≥50 h/year before their injuries ≥80% quadriceps femoris muscle strength symmetry Minimal to no knee joint effusion Full and symmetric knee ROM Able to hop without pain on each leg Initiation of a running progression	12 weeks to less than 10 months after ACLR	Duration: 5 weeks Frequency: 2 times a week Exercises included: Nordic hamstrings, standing squats progressing to tuck jumps, drop jumps, triple single leg hopping, agility drills, quadriceps strengthening exercises. Specific to SAP: A sham intervention (the athlete stood on one leg on a stable surface and performed hip flexion against a resistance band with the opposite limb). Specific to SAP + PERT: 10 sessions of perturbation training (~30 min per session).	Biomechanical testing (knee kinematic and kinetic variables, and muscle and joint contact forces)	SAP training with and without perturbation training do not meaningfully improve walking mechanics among young female athletes. Asymmetrical gait mechanics persist to a large degree until 2 years after ACLR, long after patients have achieved symmetrical strength and functional performance and have returned to sports.
[33]	Prospective randomised control trial	Compare SAP group and SAP + PERT group with respect to tibiofemoral loading, muscle forces, and the immediate before and after intervention knee kinematics and kinetics during walking.	N: 40 (20 × SAP, 20 × SAP + PERT) Male athletes Participated in level I or II sports ≥50 h/year before their injuries ≥80% quadriceps femoris muscle strength symmetry Minimal to no knee joint effusion Full and symmetric knee ROM Able to hop without pain on each leg Initiation of a running progression	3–10 months after ACLR	Duration: 5 weeks Frequency: 2 times a week Exercises included: Nordic hamstrings, standing squats progressing to tuck jumps, drop jumps, triple single leg hopping, agility drills, quadriceps strengthening exercises. Specific to SAP: Sham intervention Specific to SAP + PERT: 10 sessions of perturbation training (~30 min per session).	Biomechanical testing (knee kinematic and kinetic variables, and muscle and joint contact forces)	Neither SAP nor SAP + PERT training appears effective at altering gait mechanics in men in the short term; however, meaningful gait asymmetries mostly resolved between post-training and 2 years after ACLR regardless of the intervention group.
[34]	Randomised controlled trial	Examine the effects of a neuromuscular training program on knee proprioception in athletes who had returned to sports following ACL reconstruction.	N: 24 (12 × experimental group, 12 × control group) Male athletes 18–30 years of age Competed in basketball, soccer, volleyball, or handball at the provincial level	6–12 months after ACLR Completed conventional rehabilitation Had returned to sports	Experimental group: Duration: 8 weeks Frequency: 2–3 times a week (total sessions: 22) Exercises included: single- and double-leg squats, lunges, drop jumps, single-leg stance on an unstable surface, countermovement jumps, long jumps, and horizontal bounds. Continue with the typical routine which focused on sport-related skills.	Knee position sense tested by an isokinetic dynamometer	Athletes who participated in the neuromuscular training program exhibited better knee proprioception for their ACL-reconstructed limb, compared to athletes who did not participate in neuromuscular training (control).
[35]	Randomised controlled trial	Assess the effects of eccentric training, plyometric training, or a combination of the above two modalities, on measures of dynamic stability, psychological readiness to return to sport, and leg symmetry index in the post-ACLR rehabilitation period of elite female athletes.	N: 40 (10 × eccentric group, 10× plyometric group, 10× combined eccentric and plyometric group, 10 × control group) Female athletes Non-contact ACL injury during sporting activity Performed a systematic sports practice at the international level and were a member of the Tunisian national team in their respective sport Athletes exercise 6–8 times per week including competition	All ACLR were performed by the same 2 knee surgeons who had at least 20 years of technical experience with ACLR Received same rehabilitation protocol 2 weeks after ACLR × 12 weeks Participated in the study 14 weeks after ACLR	Duration: 6 weeks Control group: instructed to follow their traditional program Experimental groups: Frequency: 2 additional sessions per week × 60 min per session (12 sessions in total in addition to the traditional program) Exercises included: Eccentric group: Nordic hamstring, eccentric hamstring curl, quadriceps eccentric leg extension, glute-hamstring raise. Plyometric group: Standing vertical hops, countermovement jump, depth jumps, multiple two-foot hurdle jumps, two-foot jumps (forward, backward, lateral), single-foot jumps. Combined group: combination of eccentric and plyometric groups	Dynamic stability (Y-BAL) Psychological readiness to return to sport (LKS and RSI) Leg symmetry index for the single-leg hop tests	Despite all of the training methods inducing improvement outcomes to various extents, combined (eccentric/plyometric) training was the most effective protocol to stimulate positive changes in both stability and functional performance.
[36]	Randomised controlled trial	To examine the effect of a jump training program on patient-reported function and biomechanical measures and to determine whether a high-repetition program with decreased intensity via body weight support (BWS) will improve functional, mechanical, and neuromuscular outcomes.	N: 19 (9 × JTBW group, 10 × JTBWS group) Age: 12–35 Participated in recreational or competitive sports at a Tegner Activity Scale level higher than 4 One or more of the below during the assessment for eligibility: Scores lower than 75 on the IKDC An LSD lower than 75% in a single-legged hop for distance test A peak knee moment less than 2.3 body weight Lower than 80% of the non-surgical side during a single-legged landing test	Between 6–48 months post-ACLR Had been cleared for sports participation by their surgeon	Duration: 8 weeks Frequency: twice-weekly ×1 h long Form of training: Individual sessions Exercises and progression: Jump training under normal body weight conditions (JWBW) group Training progressed from 80–100 contacts per session in the first week to 120–200 contacts per session in the eighth week. Jump training augmented by a customed body weight support system group (JWBWS): Training was initiated at a BWS level of 30%, then decreased by 10% every 2 weeks, and without BWS at the final 2 weeks of training.	Patient-reported outcomes: IKDC, GRoC Performance-based functional outcomes: SLHD Biomechanical outcomes: Electromyographic testing, kinematic, and kinetic data, joint angles and moments	Both groups demonstrated significant improvements in both patient-reported and performance-based measures, while there were no significant differences between groups. However, the patients in the JTBW group had a statistically higher probability of effusion with training, which may indicate the improved training tolerance with less risk for knee effusion in the JTBWS group is clinically preferential.
[37]	Randomised controlled trial	To investigate the impact of running retraining on the muscular strength of the knee’s extensors and flexors at 4 and 6 months after ACLR.	N: 80 (21 × Patella tendon retrained group, 20 × Patella tendon control group, 19 × hamstring tendon retrained, 20 × hamstring tendon control group) Male athletes Age: 18–50 Practice of a pivot-contact sport three times a week before the ACL rupture	4–6 months post-ACLR All surgical procedures were performed by arthroscopy by two experienced surgeons Rehabilitation was standardised	The retrained group: Duration: 8 weeks Frequency: 3 times a week Exercises and progression: running intensity chosen based on the percentage of the maximal heart frequency. Control group: No intervention was given.	Knee pain (VAPS) Knee function (Lysholm Score) Knee’s clinical stability (Lachman test and the jerk, or pivot shift test) Instrumental laxity Knee’s isokinetic strength	The running retraining program did not appear to influence the knee’s muscular and functional recovery.

*SAP:* Strengthening, agility, and secondary prevention; *SAP + PERT:* Strengthening, agility, and secondary prevention plus perturbation training; *SD:* standard deviation; *ACLR:* Anterior cruciate ligament reconstruction; *IKDC:* International Knee Documentation Committee Subjective Knee Evaluation Form; *KOOS:* Knee Injury and Osteoarthritis Outcome Score; *ROM:* Range of motion; *pKFA:* peak knee flexion angle; *pKExtA:* peak knee extension angle; *Y-BAL:* Y balance test; *LKS:* Lysholm Knee Scale; *RSI:* Return to sport index; *BWS:* body weight support; *LSD:* Limb symmetry index; *JWBW:* Jump training under normal body weight conditions; *JWBWS:* Jump training augmented by a customed body weight support system; *GRoC:* The Global Rating of Change; *SLHD:* Single-legged hop for distance; *VAPS:* Visual Analogue Scale for Pain.

**Table 2 ijerph-20-04821-t002:** The risk of bias assessment of the included studies according to the Cochrane risk of bias framework [28].

	Random Sequence Generation	Allocation Concealment	Blinding of Participants and Personnel	Blinding of Outcome Assessment	Incomplete Outcome Data	Selective Outcome Reporting	Other Sources of Bias
Arundale et al., 2017 [20]			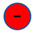		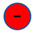		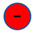
Capin et al., 2017 [30]	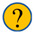	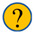	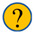	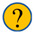	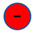		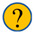
Ghaderi et al., 2021 [31]			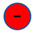	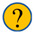			
Capin et al., 2019 [32]		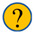	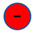				
Capin et al., 2018 [33]		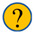	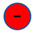		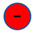		
Ghaderi et al., 2020 [34]			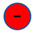				
Kasmi et al., 2021 [35]	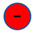	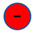	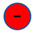	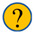			
Elias et al., 2018 [36]			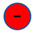				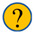
Dauty et al., 2010 [37]	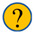	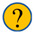	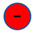	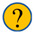			


: Low risk of bias 
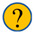
: Unclear risk of bias 
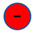
: High risk of bias.

## Data Availability

Not applicable.

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
