# Peer review of "Secondary Anterior Cruciate Ligament Injury Prevention Training in Athletes: What Is the Missing Link?"

_ijerph, 2023, doi:10.3390/ijerph20064821_

Round 1

Reviewer 1 Report

Dear Authors, intriguing manuscript, with registration PROSPERO. There is some methodological concern regarding RoB and then I would recommend concluding on the secondary prevention issue and the results of your review, limiting the need for further studies and the focus on primary prev. Thank you

9 is

14 scopus, wos?

15 these statements concern primary prevention? because on lines 18-19 you say the opposite.

21 in the light of the previous statements, is this really so?

60 “The altered neuromuscular timing and recruitment can lead to dynamic knee valgus stress in the lower limb; it also reported that female athletes show four times greater activation of their hamstring muscles than males during knee dynamic stress motion”

Ref: Marotta, N., Demeco, A., Moggio, L., Isabello, L., & Iona, T. (2020). Correlation between dynamic knee valgus and quadriceps activation time in female athletes. Journal of Physical Education and Sport, 20(5), 2508-2512 http://doi.org/10.7752/jpes.2020.0534

65-74 is a duplicate paragraph..

92 put the prospero in the abstract to increase the caliber of the manuscript at first glance..

128 lack of quality analysis, what type of risk of bias did you use.. I would recommend the Joanna Briggs Institute

Exhaustive table but for a correct drafting of a systematic review an assessment of the Risk of Bias cannot be missing.. follow the PRISMA guidelines

167 Paraphrase the study objective as the first sentence of the discussion

261 detach limitations with a space, it is necessary for the reader to have them available at a glance..

Author Response

Thank you very much for your comments. We hope that our revision will be sufficient for supporting publication.

9 is

Response: Thank you. Change as suggested.

14 scopus, wos?

Response: Authors believe that PubMed (1964 to 2021) and EBSCOhost (CINAHL, MEDLINE, ScienceDirect [1985 to 2021]) databases provides enough coverage for the search.

15 these statements concern primary prevention? because on lines 18-19 you say the opposite.

Response: Thank you for your comment. The statements has been revised as below

Line 15-19: The existing evidence is prone to suggest neuromuscular training, eccentric strengthening, and plyometric exercises may have a potential impact to improve biomechanical, functional, and psychological outcomes in athletes, however the studies on the prevention of second ACL injury in athletes is scarce and inconclusive.

21 in the light of the previous statements, is this really so?

Response: Thank you for your comment. The sentence has been removed to avoid confusion.

60 “The altered neuromuscular timing and recruitment can lead to dynamic knee valgus stress in the lower limb; it also reported that female athletes show four times greater activation of their hamstring muscles than males during knee dynamic stress motion”

Ref: Marotta, N., Demeco, A., Moggio, L., Isabello, L., & Iona, T. (2020). Correlation between dynamic knee valgus and quadriceps activation time in female athletes. Journal of Physical Education and Sport, 20(5), 2508-2512 http://doi.org/10.7752/jpes.2020.0534

Response: Thank you for the comment. The statement has been added and cited.

65-74 is a duplicate paragraph..

Response: Thank you for your comment. The sentence has been removed to avoid confusion.

92 put the prospero in the abstract to increase the caliber of the manuscript at first glance.

Response: Thank you for the great suggestion. The PROSPERO Registration number has been added to the abstract.)

128 lack of quality analysis, what type of risk of bias did you use.. I would recommend the Joanna Briggs Institute

Exhaustive table but for a correct drafting of a systematic review an assessment of the Risk of Bias cannot be missing.. follow the PRISMA guidelines

Response: Sorry for missing the important information. The risk of bias was assessed using the Cochrane risk of bias framework. The manuscript is amended as suggested.

167 Paraphrase the study objective as the first sentence of the discussion

Response: Revised as suggested.

261 detach limitations with a space, it is necessary for the reader to have them available at a glance.

Response: Revised as suggested.

Reviewer 2 Report

Results showed that neuromuscular training, eccentric strengthening, and plyometric exercises may have a positive impact on the biomechanical, functional, and psychological outcomes in athletes, while secondary prevention programmes may be ineffective in influencing gait asymmetries. So that may indicate that a primary ACL prevention programme may also be valid in secondary ACL injury prevention.

For data interpretations I can say that Table 1. (Summary of the included studies) is relatively poor. It should be more more clear (transparent) for the reader.

Author Response

Response: Thank you very much for the appreciation and kind suggestion. We have tried to simplify the table, however the key information is remaining at the moment. Understand that the current topic is a complicated issue regarding the prevention of 2nd ACL injury. We hope that the table will be accepted in the current form in order to provide sufficient information to the reader. Thank you very much for your understanding.

Round 2

Reviewer 1 Report

Dear authors, the manuscript is of a distinguished caliber, I can suggest its suitability for publication. Given the registration on PROSPERO it is a pity not to mention it in the abstract, it could give more charm to the reader to deepen the full text reading.